# Targeting FGFR Pathway Is Not an Effective Therapeutic Strategy in Patients with Unselected Metastatic Esophagogastric Cancer Resistant to Trastuzumab

**DOI:** 10.3390/jpm13030508

**Published:** 2023-03-11

**Authors:** Camilla Zecchetto, Alberto Quinzii, Simona Casalino, Marina Gaule, Camilla Pesoni, Valeria Merz, Silvia Pietrobono, Domenico Mangiameli, Martina Pasquato, Stefano Milleri, Simone Giacopuzzi, Maria Bencivenga, Anna Tomezzoli, Giovanni de Manzoni, Davide Melisi

**Affiliations:** 1Investigational Cancer Therapeutics Clinical Unit, Azienda Ospedaliera Universitaria Integrata, 37134 Verona, Italy; 2Digestive Molecular Clinical Oncology Research Unit, Università degli Studi di Verona, 37134 Verona, Italy; 3Medical Oncology Unit, Santa Chiara Hospital, 38122 Trento, Italy; 4Centro Ricerche Cliniche, 37134 Verona, Italy; 5General and Upper GI Surgery Division, Department of Surgery, Dentistry, Pediatrics and Gynaecology, University of Verona, 37134 Verona, Italy; 6Anatomical Pathology Unit, Azienda Ospedaliera Universitaria Integrata, 37126 Verona, Italy

**Keywords:** esophagogastric cancer, HER-2, trastuzumab resistance, FGFR3, pemigatinib

## Abstract

Trastuzumab plus chemotherapy is the standard of care for the first-line treatment of patients with HER2+ advanced esophagogastric (EG) cancer. Nevertheless, patients frequently develop resistance. In preclinical models, we identified the overexpression of Fibroblast Growth Factor Receptor (FGFR) 3 as a mechanism potentially involved in trastuzumab-acquired resistance. FGFR inhibition could be a potential mechanism as a second-line treatment. In this Simon’s two-stage phase 2, single arm study, patients with advanced EG cancer refractory to trastuzumab-containing therapies received pemigatinib, an inhibitor of FGFR. The primary end point was the 12-week progression-free survival rate. Translational analyses were performed on tissue and plasma samples. Eight patients were enrolled in the first stage. Although the 6-week disease control rate was 25%, only one patient achieved a stable disease after 12 weeks of treatment. The trial was discontinued before the second stage. Two out of six evaluable tumor samples expressed FGFR3. No *FGFRs* amplification was detected. *HER2* amplification was lost in three out of eight patients. Three patients had an high Tumor Mutational Burden, and two of them are significantly long-term survivors. These results do not support the therapeutic efficacy of targeting FGFR in unselected patients with advanced EG cancer, who are refractory to trastuzumab-containing therapies.

## 1. Introduction

Esophagogastric (EG) cancer constitutes a major global health problem, as it remains the fifth most frequently diagnosed cancer worldwide and the third leading cause of cancer-related deaths [1]. EG cancer patients frequently present with unresectable locally advanced or metastatic disease and, therefore, are candidates for systemic treatments.

Despite the efforts in identifying novel molecularly targeted approaches [2], only three targeted therapeutic agents—trastuzumab, ramucirumab, and pembrolizumab—have been approved for the treatment of patients with advanced EG cancer. In particular, trastuzumab—the monoclonal antibody directed against the extracellular domain of the human epidermal growth factor receptor 2 (HER2)—is still the only treatment that is effective against the altered product of an oncogene, the erythroblastic leukemia viral oncogene homolog 2 (*ERB-B2*), which is amplified in a range from 12 to 23% in EG cancer patients. This agent has been approved for the first-line treatment of patients with HER2-overexpressing metastatic esophagogastric junction (EGJ)/gastric cancer (GC) in combination with chemotherapy [3].

Patients with HER2 overexpressing tumors usually respond to upfront trastuzumab, but after a certain period of therapy, almost all patients develop resistance. The selection of HER2 not amplified clones is a frequent mechanism of trastuzumab resistance. Beyond these, other additional mutations in receptor tyrosine kinases, RAt Sarcoma (RAS) and PhosphatidylInositol 3-Kinase (PI3K) pathways are emerging as relevant mechanisms of intrinsic and acquired resistance to anti-HER2 therapeutic strategies. Patients with these co-alterations have a lower benefit from trastuzumab therapy and a shorter progression-free survival (PFS) [4,5].

The Fibroblast Growth Factor Receptor (FGFR) family consists of four tyrosine kinases transmembrane receptors (FGFR1-4). The binding of a ligand to FGFRs triggers the activation of downstream RAS/Rapidly Accelerated Fibrosarcoma (RAF)/Mitogen-activated protein kinase (MEK), Janus Kinase (JAK)/ Signal Transducer and Activator of Transcription (STAT) and PI3K/Ak strain Transforming (AKT) pathways [6]. Our group has recently shown that in EG cancer models, the overexpression of FGFR3 and Fibroblast Growth Factor 9 (FGF9) sustains acquired resistance to trastuzumab, through the activation of PI3K/AKT/Mammalian target of the rapamycin (mTOR) signaling pathway. In vivo, the treatment of trastuzumab-resistant murine models with a selective inhibitor of FGFR3 was associated with a reduction in tumor burden and an increase in overall survival (OS) compared to control [7].

This preclinical evidence led us to design the FiGhTeR trial, a Simon’s two-stage phase 2, single arm, open-label study aiming to assess safety, tolerability and activity of pemigatinib (INCB054828, Pemazyre^®^, Incyte Co., Wilmington, DE, USA), a potent and selective oral inhibitor of FGFR1, 2 and 3 [8] in patients with metastatic EG cancer resistant to first-line trastuzumab-containing therapies [9]. Genomic analysis of tissue and plasma samples were collected before treatment and at disease progression to uncover potential determinants of sensitivity and resistance to therapy.

## 2. Materials and Methods

The FiGhTeR trial was a phase 2, single arm, open-label study to assess safety, tolerability and activity of pemigatinib as a second-line therapy approach in metastatic EG cancer patients resistant to first-line trastuzumab-containing therapies. It was a single-institution trial conducted at the early phase clinical trial unit Centro Ricerche Cliniche at the University Hospital of Verona, Italy.

### 2.1. Patients

Main eligibility criteria were age ≥ 18 years, histologically confirmed advanced or metastatic EG cancer, disease progression within three months of the last dose of first-line trastuzumab-containing therapy, fresh biopsy availability, measurable disease per Response Evaluation Criteria in Solid Tumors (RECIST) version 1.1, Eastern Cooperative Oncology Group (ECOG) performance status 0/1, life expectancy ≥ 3 months, and adequate bone marrow, liver and renal function. Exclusion criteria included corneal disorders, serious cardiovascular disease and elevated serum phosphate and calcium levels.

### 2.2. Study Treatment

All patients received self-administered, oral pemigatinib at a dose of 13.5 mg once daily (21-day cycle; 2-weeks-on, 1-week-off), until radiologic disease progression, unacceptable toxicity, withdrawal of consent or patient/physician choice.

### 2.3. End Points and Assessment

The primary end point was the 12-week PFS rate, defined as the probability of being without progression (loco-regional or distant) or death due to any cause at 12 weeks from trial enrollment. The assessment of anti-tumour efficacy was carried out by performing a CT or MRI scan every 6 weeks for the first 4 cycles and subsequently every 9 weeks.

The secondary end points included overall survival (OS), disease control rate (DCR) and safety profile (incidence of ≥grade 3 adverse events (AEs) using Common Terminology Criteria for Adverse Events—CTCAE—version 4.0). The medical records of the patients and the electronic case report forms (eCRFs) both contain complete reports of all AEs. During the course of the study’s treatment period, any significant adverse events (SAEs) were to be reported within 24 h. The safety analysis set included all patients who received study treatment.

Other exploratory objectives included collecting and banking serial blood and tumor tissue specimens from subjects at baseline and at the time of pemigatinib progression for correlative biomarker studies.

### 2.4. Statistical Design and Sample Size

A Simon’s two-stage design was used for conducting the trial [10]. The null hypothesis for the 12-week PFS rate was 0.2, and the alternative hypothesis was 0.5. The trial was carried out in two stages. In stage I, a total number of 8 patients were accrued. If there were 2 or fewer responses among these 8 patients, the study would have been stopped early. Otherwise, an additional 10 patients would have been accrued in stage II, resulting in a total number sample size of 18. If there had been 7 or more responses among these 18 patients, we would have rejected the null hypothesis and claimed that the treatment was promising. The design controlled the type I error rate at 0.05 and yielded the power of 0.8.

### 2.5. Translational Research Analyses

Tissue and plasma samples were collected at enrollment and after progression under treatment with trastuzumab (Figure 1). The expression of HER2 and FGFR1-3 was measured by immunohistochemistry (IHC). Briefly, formalin-fixed paraffin-embedded (FFPE) 4 μm tissue sections were deparaffinated with xylene, and antigen retrieval was performed in ethylene-diamine-tetra-acetic acid buffer. Following antigen retrieval, slides were incubated overnight at 4 °C according to the manufacturer’s instructions with the following rabbit anti-human primary antibodies: anti-FGFR1 (M2F12, sc-57132, 1:100; Santa Cruz Biotechnology INC., Dallas, TX, USA), anti-FGFR2 (HPA035305, 1:100; Atlas Antibodies, Bromma, Sweden) and anti-FGFR3 (sc-13121, 1:100; Santa Cruz Biotechnology INC., Dallas, TX, USA). Slides were washed in Tris-buffered saline buffer and then incubated for 30 min with the appropriate horseradish peroxidase-conjugated secondary antibody. Color was detected with an HRP Anti-Rabbit IgG Polymer Detection Kit (Novocastra, Leica Microsystems, Wetzlar, Germany) for human tissue. The slides were counterstained with Meyer’s hematoxylin (Peroxidase Detection System; Leica Microsystems Inc., Wetzlar, Germany). To ensure antibody specificity, consecutive sections were incubated with isotype-matched control immunoglobulins and in the absence of the primary antibody. In these cases, no specific immunostaining was detected. The expression of proteins was detected as membrane, cytoplasmic or nuclear brown staining of varying intensity in neoplastic cells. The slides were evaluated independently using light microscopy by two pathologists who were blinded to the treatments.

Genetic alterations were identified in tissue or circulating tumor-free DNA by performing next generation sequencing (NGS) through TruSight Oncology 500 for tumor tissue assay and TruSight Oncology 500 ctDNA for liquid assay. This platform is NGS-based, hybrid-capture assays that enable Comprehensive Genomic Profiling (CGP) through analysis of key biomarkers. The platform employs whole-genome shotgun library construction and hybridization-based capture of DNA extracted from FFPE tumor tissue prior to uniform and deep sequencing on the Illumina^®^ NextSeq550Dxa. Sequence data are subsequently processed using a customized analysis pipeline designed to detect all classes of genomic alterations, including substitutions, insertions and deletions; copy number alterations and select rearrangements. No analyses were performed on matched normal tissue samples. The typical median depth of coverage is >500×, with >99% of exons at coverage >100×. Liquid assay employs a single DNA extraction method to obtain circulating cell-free DNA (cfDNA) from plasma derived from anti-coagulated peripheral whole blood. Extracted ctDNA undergoes whole-genome shotgun library construction and hybridization-based capture of 523 cancer-related genes. Hybrid-capture selected libraries were sequenced with deep coverage using the NovaSeq^®^ 6000 platform, and sequence data were processed using a custom analysis pipeline designed to detect genomic alterations.

## 3. Results

### 3.1. Patient Characteristics

Between November 2019 and February 2021, eight patients were enrolled in the first stage of the trial. Patients’ and tumor characteristics are detailed in Table 1. Patients’ median age was 65 years, and the majority of them were in good clinical condition (75% with PS ECOG = 0, 25% with PS ECOG = 1).

### 3.2. Treatment Compliance and Safety

Patients’ compliance during the treatment was 100%. No AEs led to study discontinuation or interruption. The most common AEs included hyperphosphatemia (*n* = 7, 87.5%), fatigue (*n* = 3, 37.5%), diarrhea (*n* = 2, 25%), mucositis (*n* = 1, 12.5%) and hypercreatininemia (*n* = 1, 12.5%). These AEs were reported during the period of drug assumption. All of these AEs were reported to be grade 1–2 (Table 2).

### 3.3. Activity

After the first tumor evaluation at 6 weeks, out of 8 patients 1 exhibited a partial response (PR) (12.5%) and 1 stable disease (SD) (12.5%), whereas 6 patients (75%) had a progressive disease (PD), accounting for a 25% DCR (2/8). Notwithstanding the measurable DCR, the primary end point of a 12-week PFS rate was not met since after 12 weeks of treatment only 1 patient achieved SD (Figure 2).

Median PFS was 2.1 months (95% CI 1.10–3.10), with the longest PFS duration (5.0 months) measured in the patient achieving a PR. During the median follow-up of 28.3 months, the median OS was 8.2 months (95% CI 4.86–11.54); two patients are still alive and are currently receiving a third-line chemotherapeutic treatment (Table 3).

### 3.4. Translational Analyses

Available tissue or plasma samples were processed to identify potential biomarkers of response to pemigatinib or other relevant genomic findings.

Tumor tissue samples collected at the time of enrollment from 6 out of 8 patients met minimum quality control metrics for processing and were analyzed by IHC for the expression of FGFR 1–3 receptors. We measured a weak to moderate expression of FGFR3 in two patients and a weak expression of FGFR1 in one patient.

Moreover, we performed NGS analysis on tissue and plasma samples collected at the time of enrollment. Four out of eight tumor tissue samples yielded inadequate tumor content for genomic analysis, whereas plasma circulating tumor DNA (ctDNA) met minimum quality control metrics for processing in all of the samples.

In those patients where both samples were available, we measured a good concordance between the genomic alterations identified in tumor tissue and plasma (Table 4). Interestingly, NGS analysis conducted on plasma samples reported a gene copy number of HER2 generally lower than did the analysis conducted on tissue samples. Nonetheless, we measured the loss of *HER2* amplification in three out of eight patients.

Moreover, we detected a single-nucleotide polymorphism of FGFR4 p.G388R 162G>A in four out of eight patients and KRAS mutation or amplification in two patients out of eight. Other recurrent mutations were Tumor Protein p53 (TP53) (p.P72R c.215C>G) and ASXL Transcriptional Regulator 1 (ASXL1) (p.L815P c.2444T>C) in 87.5% of patients, Cyclin D3 (CCND3) (p.S259A c.775T>G) in 75% of patients, BReast CAncer gene (BRCA1) (p.K1183R c.3548A>G) in 62.5% of patients, Kinase Insert Domain Receptor (KDR) (p.Q472H c.1416A>T) and tyrosine-protein kinase Kit (KIT) (p.M541L c.1621A>C) in 37.5% of patients.

Worthy of note, high Tumor Mutational Burden (TMB) was detected in 3 patients (37.4 muts/Mb, 25.5 muts/Mb and 20 muts/Mb, respectively), and 2 are still alive at 4 and 6 years since diagnosis of metastatic disease. All these results are detailed in Table 4 and Table 5.

## 4. Discussion

The aim of the trial was to assess the safety, tolerability and activity of the FGFR inhibitor pemigatinib as a second-line treatment in patients affected by HER2-positive metastatic EG cancer at diagnosis, who had become resistant to first-line trastuzumab-containing therapies. Our results suggest that the single targeting of the FGFR pathway is not an effective therapeutic strategy in this clinical setting.

In the present trial, EG cancer patients were not prescreened for FGFRs genomic alterations at enrollment. Genomic alterations in gene coding for FGFRs are rare in EG cancers, ranging between 3 to 10% of the cases in different series and mainly represented by amplifications or rearrangements of the FGFR2 gene [11]. According to the Cancer Genome Atlas Research Network molecular classification data, 9% of chromosomal instability (CIN) and 8% of genomically stable (GS) gastric cancer subtypes harbored amplification of *FGFR2* [2], while esophageal adenocarcinoma had amplification of *FGFR1* or *FGFR2* in 4% and 3% of the cases, respectively [12]. A more recent comprehensive genomic profiling of 6667 individual EG adenocarcinoma samples identified a total of 269 (4.0%) *FGFR2*-altered cases, consisting of 193 *FGFR2* amplification (2.9%), and 23 FGFR2 rearrangement (0.3%) [13]. Amplification of *FGFR2* could also be detected in ctDNA in 7.6% of cases, including a subset of patients with extremely high-level ctDNA amplifications [14]. Most importantly, these studies conducted by analyzing samples from untreated patients demonstrated that amplifications of *FGFR2* are mutually exclusive with those of other genes coding for tyrosine-kinase receptors, such as ERBB2, epidermal growth factor receptor (EGFR) or Mesenchymal Epithelial Transition (MET) [15]. Moreover, targeted sequencing in plasma and tissue biopsy samples from HER2+ patients to track the resistance during trastuzumab treatment did not measure any enrichment in FGFRs genomic alterations [4,16]. Such evidence and the model emerging from our preclinical studies [7] supported the hypothesis that the resistance to trastuzumab could not be sustained by pre-existing or acquired amplifications or rearrangements of FGFR pathway, but by a dynamic transcriptional overexpression of FGFR3. According to this hypothesis, EG cancer patients were considered for enrollment based on their progression under trastuzumab-containing regimens and were not prescreened for FGFRs genomic alterations. In this first-step cohort, NGS analysis on tissue or ctDNA did not report any pathogenetic somatic alteration of FGFRs genes. Worthy of note, half of the patients had a germline variant of *FGFR4* p.G388R c.1162G>A, a commonly occurring single-nucleotide polymorphism which has been associated with several cancers. The relative amino-acid change enhances the recruitment and activation of signal transducer and activator of transcription 3 (STAT3), and, thus, accelerates cancer progression [17]. In six out of eight patients, tumor tissue samples met minimum quality control metrics for IHC analysis. A weak to moderate expression of FGFR3 was measured in two patients, though neither of them responded to FGFR inhibition.

The activity of pemigatinib as second-line treatment in FGFR wild-type patients with initially HER2-positive EG cancers who progressed under a trastuzumab-containing first-line chemotherapy regimen was not sufficient to meet the hypothesis of the trial’s design. At 6 weeks, we measured a PR in 1 (12.5%), an SD in 1 (12.5%) and a PD in 6 (75%) out of 8 patients, accounting for a relevant DCR of 25%. However, only 1 patient maintained a stable disease after 12 weeks of treatment; thus, the trial was discontinued before the second stage.

As reported for pemigatinib in other clinical settings, such as metastatic cholangiocarcinoma [18], hyperphosphatemia was the most frequent adverse event. Hyperphosphatemia events were of low severity and did not require dose reductions or interruptions of treatment.

In this study, NGS analysis was performed on tissue and plasma samples collected at the time of enrollment to identify potential determinants of sensitivity and resistance to treatment. Genomic characterization of tissue samples in patients with advanced EG cancer is generically affected by logistic challenges associated with a sufficient sampling and adequate tumor content. In our study, half of the tissue samples yielded inadequate tumor content for genomic analysis, whereas all of the plasma samples met minimum quality control metrics for processing, and the genomic alterations identified in ctDNA were largely concordant with those in tissue samples. Comparable efficiency in detecting alterations, potential to overcome tumor heterogeneity by analyzing DNA released by all the different primary and metastatic lesions and the possibility to spare patients a new biopsy procedure are relevant advantages supporting the use of ctDNA sequencing in EG cancer patients who may be candidates for targeted treatments. Notably, these sequencing analyses confirmed the loss of *HER2* amplification in three out of eight patients. Eradication of HER2-expressing cells by trastuzumab could result in the emergence of HER2-negative clones, as previously reported in approximately 30% of patients with initially HER2+ EG cancers who received trastuzumab-containing first-line chemotherapy regimens [4,19].

A high TMB, defined as the total number of somatic genomic alterations per million bases, has been recently associated with a superior survival rate in EG cancer patients in the cancer genome atlas database [20]. In our study, 3 out of 8 patients had a high TMB with >20 muts/Mb. These patients had a uniquely long-term survival with 2 of them still alive at 4 and 6 years since diagnosis of metastatic disease. This evidence seems to corroborate the favorable prognosis associated with high TMB.

The FGFR signaling pathway remains one of the most promising therapeutic targets in the treatment of EG cancer. Therapy with bemarituzumab, a humanized IgG1 monoclonal antibody selective for FGFR2b, in combination with chemotherapy was found to improve survival in patients with FGFR2b-positive, HER2-negative frontline advanced EG cancers [21]. Nonetheless, early clinical experiences with ATP-competitive FGFR kinase inhibitors showed how only 5% of EG cancer patients, presenting an homogeneously high-level clonal *FGFR2*-amplification might achieve a clinical benefit from such an approach [22,23].

In conclusion, these data do not support the therapeutic efficacy of single agents targeting FGFR in unselected patients with advanced EG cancer refractory to trastuzumab-containing therapies. Additional research is needed to clarify the mechanisms of acquired resistance to anti-HER2 therapeutic strategies, and to identify novel biomarkers for patients’ selection and targeted treatments.

## Figures and Tables

**Figure 1 jpm-13-00508-f001:**
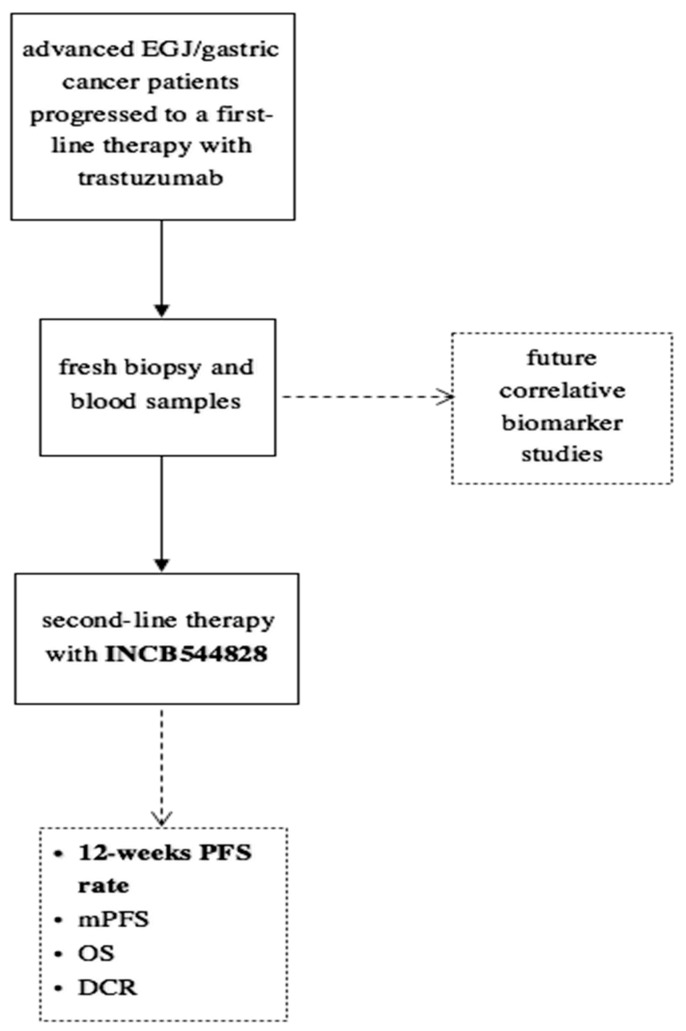
Study design.

**Figure 2 jpm-13-00508-f002:**
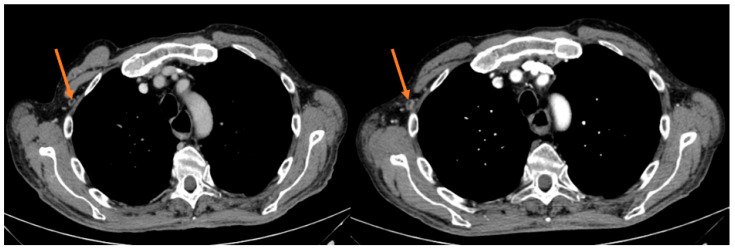
Intercostal lesion before and after 12 weeks of treatment in the responder patient.

**Table 1 jpm-13-00508-t001:** Patients and tumor characteristics.

Characteristics	*n* = 8
Gender	
Male	7 (88%)
Female	1 (12%)
Age, median (range)	65.0 (CI 95% 51.6–78.4)
Performance status (ECOG)	
0	6 (75%)
1	2 (25%)

**Table 2 jpm-13-00508-t002:** Treatment-related adverse events.

Adverse Events	*n* = 8 Grade 1–2	Grade 3
Any	8 (100%)	0
Hyperphosphataemia	7 (87.5%)	0
Fatigue	3 (37.5%)	0
Diarrhea	2 (25%)	0
Mucositis	1 (12.5%)	0
Hypercreatininemia	1 (12.5%)	0

**Table 3 jpm-13-00508-t003:** Antitumor activity.

Category	*n* = 8	% (95% CI)
Best overall response at first CT-scan		
Partial response	1	12.50%
Stable disease	1	12.50%
Progression disease	6	75
12-week PFS rate		
Stable disease	1	12.50%
Progression disease	7	87.50%
Progression-Free Survival (months)	2.1	1.1–3.1
Disease control rate	1	12.50%
Median Overall Survival (months)	8.2	4.9–11.5

**Table 4 jpm-13-00508-t004:** Translational analyses.

Patient ID	IHC HER2/AMP	IHC FGFR1-3	NGS Solid	NGS Liquid
001	3+/−	1+, 2−, 3−	Insufficient tumor sample	TMB medium MSI stable *KRAS* 5 copies *TP53* p.P72R c.215C>G *BRCA1* p.K1183R c.3548A>G *CCND3* p.S259A c.775T>G *RICTOR* 2 copies *PDGFRA* p.G79D c.236G>A *TP53* p.D281N c841G>A
002	3+/+	1−, 2−, 3+	TMB medium MSI stable *ERRB2* 39 copies *FGFR4* p.G388R c.1162G>A *KDR* p.Q472H c.1416A>T *TP53* p.G266E c.797G>A *KRAS* p.Q61H c.183A>C *ATM* p.5707P c.2119T>C	TMB medium MSI stable *ERRB2* 9 copies *FGFR4* p.G388R c.1162G>A *KDR* p.Q472H c.1416A>T *TP53* p.P72R c.215C>G *ASXL1* p.L815P c.2444T>C *CCND3* p.S259A c.775T>G *TP53* p.G266E c.797G>A *KRAS* p.Q61H c.183A>C *ATM* p.5707P c.2119T>C *PIK3CA* p.E726K c.2176G>A
003	3+/+	1−, 2−, 3++	TMB medium MSI stable *ERRB2* 73 copies *EGFR 3* copies *TP53* p.P72R c.215C>G *PALB2* p.L939W c.2816T>G	TMB high (37.4 muts/Mb) MSI stable *ERBB2* 21 copies *ASXL1* p.L815P c.2444T>C *TP53* p.P72R c.215C>G *PALB2* p.L939W c.2816T>G *BRCA1* p.K1183R c.2816T>G *PPM1D* p.S468* c.1403C>G *GNAQ* p.Y101*c.303C>A *CHEK2* p.Y488*c.1464C>A
004	3+/−	1−, 2−, 3−	TMB low MSI stable CDK4 3 copy number *FGFR4* p.G388R c.1162G>A *BRCA1* p.K1183R c.3548A>G *APC* p.I1307K c.3920T>A *CDKN2A* p.H83D c.247C>G *TP53* p.L130H c.389T>A *TP53* p.R273P c.818G>C	TMB low MSI stable *FGFR4* p.G388R c.1162G>A *BRCA1* p.K1183R c.3548A>G *APC* p.I1307K c.3920T>A *ASXL1* p.L815P c.2444T>C *CCND3* p.5259° c.775T>G
005	3+/−	1−, 2−, 3−	Insufficient tumor sample	TMB low MSI stable *BRCA1* p.K1183R c.3548A>G *TP53* p.P72R c.215C>G *FGFR4* p.G388R c.1162G>A *KDR* p.Q472H c.1416A>T *ASXL1* p.L815P c.2444T>C *CCND3* p.S259A c.775T>G *SLX4* p.N1834S c.5501A>G *DNM3A* p.E616 c.1846G>T
006	3+/−	1−, 2−, 3−	TMB low MSI stable *ERBB2* 11 copies *TP53* p.C275Vfs c.823delT *TP53* p.P72R c.215C>G *KDR* p.Q472H c.1416A>T *PIK3CA* p.R93W c.277C>T	TMB low MSI stable *ERBB2* 3 copies *TP53* p.C275Vfs c.823delT *TP53* p.P72R c.215C>G *KDR* p.Q472H c.1416A>T *PIK3CA* p.R93W c.277C>T *ASXL1* p.L815P c.2444T>C *KIT* p.M541L c.1621A>C *SMC1A* p.R711Q c.2132G>A *DNMT3A* p.R598 c.1792C>T
007	3+/−	Insufficient tumor sample	Insufficient tumor sample	TMB high 25.5 Muts/Mb MSI stable *ERBB2* 6 copies *TP53* p.P72R c.215C>G *BRCA1* p.K1183R c.3548A>G *ASXL1* p.L815P c.2444T>C *KIT* p.M541L c.1621A>C *CCND3* p.S259A c.775T>G *TP53* p.G244D c.731G>A *TP53* p.C275Y c.824G>A *TP53* p.Y220C c659A>G *CREBBP* p.A981T c.2941G>A
008	3+/−	Insufficient tumor sample	Insufficient tumor sample	TMB high 20.0 Muts/Mb MSI stable *ERBB2* 2 copies *TP53* p.P72R c.215C>G *BRCA1* p.K1183R c.3548A>G *FGFR4* p.G388R c.1162G>A *ASXL1* p.L815P c.2444T>C *KIT* p.M541L c.1621A>C *CCND3* p.S259A c.775T>G *TP53* p.C238Y c.713G>A *TP53* p.R158H c473G>A *CHEK2* p.R180Q c.539G>A

**Table 5 jpm-13-00508-t005:** High-frequency genomic mutations.

Genomic Alterations	*n* = 8
*TP53* p.P72R c.215C>G	7 (87.5%)
*ASXL1* p.L815P c.2444T>C	7 (87.5%)
*CCND3* p.S259A c.775T>G	6 (75%)
*BRCA1* p.K1183R c.3548A>G	5 (62.5%)
*FGFR4* p.G388R c.1162G>A	4 (50%)
*KDR* p.Q472H c.1416A>T	3 (37.5%)
*KIT* p.M541L c.1621A>C	3 (37.5%)

## Data Availability

Data is unavailable due to privacy or ethical restrictions.

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
