# Peer review of "Targeting FGFR Pathway Is Not an Effective Therapeutic Strategy in Patients with Unselected Metastatic Esophagogastric Cancer Resistant to Trastuzumab"

_jpm, 2023, doi:10.3390/jpm13030508_

Round 1

Reviewer 1 Report

REVIEW COMMENTS

Journal: Journal of Personalized Medicine

Title: Targeting FGFR pathway is not an effective therapeutic strategy in patients with unselected metastatic esophagogastric cancer resistant to trastuzumab 

The current manuscript involves a patient study using Simon’s two-stage design in advanced or metastatic esophagogastric cancer patients who underwent first-line trastuzumab therapy. 8 patients were recruited for this study, and these patients were treated with oral pemigatinib, an FGFR inhibitor, for a course of 12 weeks. Immunohistochemistry, next generation sequencing studies were conducted to evaluate the HER2 and FGFR expression, and to observe genetic alterations after pemigatinib therapy. The study concluded that the FGFR inhibitor treatment is not effective. Overall, the studies performed in the manuscript are relevant, however, some major revisions should be made, and the grammar and sentence formation need to be improved to improve the overall quality of the manuscript.

Major revisions:

1.    Abbreviations should be provided for EGJ, GC, RAS, PI3K, RAF/MEK, JAK/STAT, AKT, FGF9, CGP, FFPE, etc.

2.    The statement “Our group has recently showed that in EG cancer models the overexpression of FGFR3 and FGF9 sustains acquired resistance to trastuzumab, through the activation of PI3K/AKT/mTOR signaling pathway.” requires reference(s).

 3.    Materials and methods: In “End points and assessment progression-free survival (PFS) should be replaced with PFS, as the abbreviation is included previous paragraphs.

4.    CT/MRI scans were mentioned in the manuscript. Please provide representative examples for the scans.

5.    IHC method indicates overnight incubation with primary antibodies. Please include tissue processing prior to the treatment with antibodies such as deparaffinization, antigen retrieval, etc.

6.    IHC was performed on patient samples. Please provide representative images for this data.

7.    “Extracted ctDNA undergoes whole-genome…” ctDNA and cfDNA terms were used several times. Are these terms used alternatively or both terms exclusive?  

Minor Revisions:

Sentence restructuring and spell check is needed for some of the statements. Some of the examples are included below.

a.    Introduction, paragraph 2, line 1: “Despite the efforts in identifying dysregulated pathways and mutated genes potentially candidate for novel molecularly targeted approaches,..” The statement is difficult to comprehend. Please restructure the sentence.

b.    resistence spell check.

c.     In vivo should be italicized.

d.    “TruSight Oncology 500 for tumor tissue ass..” should be changed to “TruSight Oncology 500 for tumor tissue assay..”

Author Response

Dear Reviewer

We thank you for your many insightful and constructive comments. You have raised multiple valid and important issues that we have taken to heart and we have carefully revised the manuscript accordingly.

These changes are described in our point-by-point response on the following pages.

Thank you very much for considering our revised manuscript.  We believe that the reviewes’ excellent suggestions were very valuable in strengthening the manuscript. We hope that it is now acceptable for publication.

Major Review

Point 1: Abbreviations should be provided for EGJ, GC, RAS, PI3K, RAF/MEK, JAK/STAT, AKT, FGF9, CGP, FFPE, etc.

Response 1: Thank you for this comment. As requested, those abbreviations have been provided along the manuscript

Point 2: The statement “Our group has recently showed that in EG cancer models the overexpression of FGFR3 and FGF9 sustains acquired resistance to trastuzumab, through the activation of PI3K/AKT/mTOR signaling pathway.” requires reference(s).

Response 2: Reference is number 8

Point 3: Materials and methods: In “End points and assessment” progression-free survival (PFS) should be replaced with PFS, as the abbreviation is included previous paragraphs.

Response 3: Thank you for this comment. We replaced it.

Point 4: CT/MRI scans were mentioned in the manuscript. Please provide representative examples for the scans.

Response 4: Thank you for this comment. Now we reported the scans before and after treatment of the responder patient

Point 5: IHC method indicates overnight incubation with primary antibodies. Please include tissue processing prior to the treatment with antibodies such as deparaffinization, antigen retrieval, etc.

Response 5: As requested, we detailed the method.

Point 6: IHC was performed on patient samples. Please provide representative images for this data.

Response 6: The immunohistochemistry images are not immediately available, we are retrieving them, we ask if an extension of the terms is possible

Point 7: “Extracted ctDNA undergoes whole-genome…” ctDNA and cfDNA terms were used several times. Are these terms used alternatively or both terms exclusive?  

Response 7: Thank you for this comment. cfDNA was used only one time, during the description of NGS analysis. From cfDNA were extracted and analyzed ctDNA.

Minor Revisions:

Point 1: Sentence restructuring and spell check is needed for some of the statements. Some of the examples are included below.

  1. Introduction, paragraph 2, line 1: “Despite the efforts in identifying dysregulated pathways and mutated genes potentially candidate for novel molecularly targeted approaches,..” The statement is difficult to comprehend. Please restructure the sentence.

Response 1: Thanks, we corrected it.

Point 2: Resistence spell check.

Response 2: Thanks, we corrected it.

Point 3: In vivo should be italicized.

Response 3: Thanks, we corrected it.

Point 4: “TruSight Oncology 500 for tumor tissue ass..” should be changed to “TruSight Oncology 500 for tumor tissue assay.

Response 4: Thanks, we corrected it.

Please see the attachement for reviewed manscript

We look forward to your evaluation of our work. 

Reviewer 2 Report

Dear editor, 

I am pleased to review this interesting manuscript, which is a protocol entitled with targeting FGFR pathway is not an effective therapeutic strategy in patients with unselected metastatic esophagogastric cancer resistant to trastuzumab. The result is interesting, but I have several concerns.

1.  How could you ensured the completeness of  self-administered medication?

2. What do you mean by "esophagogastric cancer?" Is this term contains esophageal cancer? If it did, then Lauren's classification is inappropriate since it only apply to gastric cancer. 

3. What's your criteria for response assessment ?RECIST criteria? or other? Please specify it!

Author Response

Dear Reviewer,

We thank you for your many insightful and constructive comments. You have raised multiple valid and important issues that we have taken to heart and we have carefully revised the manuscript accordingly.

These changes are described in our point-by-point response on the following pages.

Thank you very much for considering our revised manuscript.  We believe that your excellent suggestions were very valuable in strengthening the manuscript. We hope that it is now acceptable for publication.

Point 1: How could you ensured the completeness of  self-administered medication?

Response 1: Patient compliance was verified by counting the tablets of the oral drug, at the beginning and at the end of each course of treatment. The count has always been reported in the patient’s medical records

Point 2: What do you mean by "esophagogastric cancer?" Is this term contains esophageal cancer? If it did, then Lauren's classification is inappropriate since it only apply to gastric cancer. 

Response 2: Patient 002 had a cancer of the gastroesophageal junction, in the lower portion, the histological report showed an adenocarcinoma. So to be fair, as suggested, we eliminate Lauren’s classification.

Point 3: What's your criteria for response assessment? RECIST criteria? or other? Please specify it!

Response 3: For the assessment we used RECIST criteria v1.1 and it is specified in materials and methods section

We look forward to your evaluation of our work. 

Please see the attachement for reviewed manuscript

Round 2

Reviewer 1 Report

The authors responded to all the comments.

Reviewer 2 Report

The author had correctly reply to all of my comment!